# Not All Diffusion Model Activations Have Been Evaluated as Discriminative Features

**Benyuan Meng**[1,2]   **Qianqian Xu**[3,4*]   **Zitai Wang**[3]
**Xiaochun Cao**[5]   **Qingming Huang**[6,3,7*]

[1]Institute of Information Engineering, CAS
[2]School of Cyber Security, University of Chinese Academy of Sciences
[3]Key Lab. of Intelligent Information Processing, Institute of Computing Technology, CAS
[4]Peng Cheng Laboratory
[5]School of Cyber Science and Tech., Shenzhen Campus of Sun Yat-sen University
[6]School of Computer Science and Tech., University of Chinese Academy of Sciences
[7]Key Laboratory of Big Data Mining and Knowledge Management, CAS
`mengbenyuan@iie.ac.cn`  `{xuqianqian,wangzitai}@ict.ac.cn`
`caoxiaochun@mail.sysu.edu.cn`  `qmhuang@ucas.ac.cn`

## Abstract

Diffusion models are initially designed for image generation. Recent research shows that the internal signals within their backbones, named activations, can also serve as dense features for various discriminative tasks such as semantic segmentation. Given numerous activations, selecting a small yet effective subset poses a fundamental problem. To this end, the early study of this field performs a large-scale quantitative comparison of the discriminative ability of the activations. However, we find that many potential activations have not been evaluated, such as the queries and keys used to compute attention scores. Moreover, recent advancements in diffusion architectures bring many new activations, such as those within embedded ViT modules. Both combined, activation selection remains unresolved but overlooked. To tackle this issue, this paper takes a further step with a much broader range of activations evaluated. Considering the significant increase in activations, a full-scale quantitative comparison is no longer operational. Instead, we seek to understand the properties of these activations, such that the activations that are clearly inferior can be filtered out in advance via simple qualitative evaluation. After careful analysis, we discover three properties universal among diffusion models, enabling this study to go beyond specific models. On top of this, we present effective feature selection solutions for several popular diffusion models. Finally, the experiments across multiple discriminative tasks validate the superiority of our method over the SOTA competitors. Our code is available at this url.

## 1 Introduction

Diffusion models [21, 10, 37, 36] are powerful generative models that progressively reconstruct images from Gaussian noises through a series of denoising steps. Typically, a U-Net [38] is trained as the noise predictor backbone to perform denoising. Recently, the impressive generative capability inspires the application to discriminative tasks such as semantic segmentation [52, 58] or semantic correspondence [29, 56]. In this direction, *diffusion feature* is one simple yet effective approach, where the intermediate signals, named activations, are extracted from the pre-trained diffusion U-Net as dense features [2, 58, 56, 29, 53, 13, 14].

---

[*]Corresponding authors.

38th Conference on Neural Information Processing Systems (NeurIPS 2024).

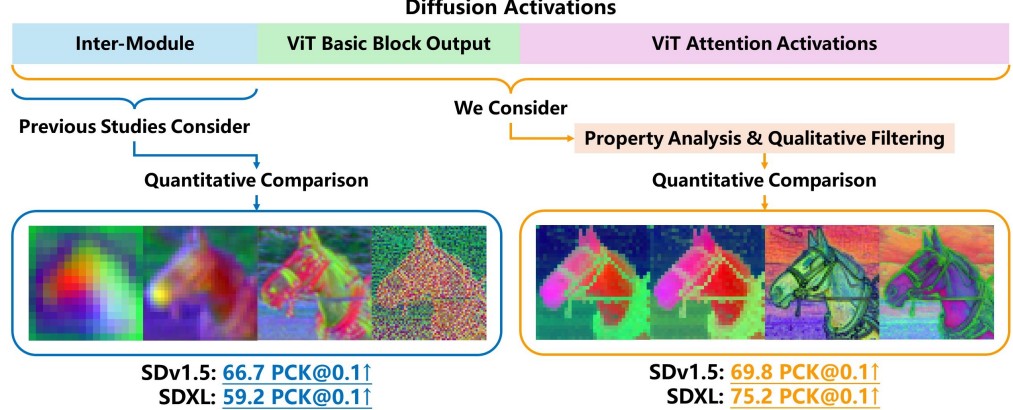

Figure 1: Prior arts only consider a small fraction of potential activations in diffusion models. As a result, more advanced diffusion architecture fails to achieve better performance (SDXL *v.s.* SDv1.5). In contrast, we consider a broader range of candidate activations. To facilitate the quantitative comparison, we first make a comprehensive and generalizable analysis to qualitatively filter out many candidates in advance. On top of this, our method achieves superior performance (75.2 PCK@0.1).

The complex architecture of the diffusion U-Net provides many activations that can serve as features. However, these activations inherently vary in quality, inducing significant performance gaps on discrimination. Hence, selecting a small yet effective subset from these activations has become a fundamental problem. In the early stage, Baranchuk *et. al* [2] perform a large-scale quantitative comparison among activations within Guided Diffusion [10]. Later, the activations they select are followed by most studies in this field, pursuing other improvements [29, 56, 58, 52, 60, 53, 24, 27].

However, we find that this fundamental issue is far from solved. On one hand, Baranchuk *et. al* [2] only consider activations between neighboring modules that comprise the main residuals. This means that many potential activations are excluded from the candidate pool, such as the queries and keys in the self-attention blocks. Moreover, recent developments of diffusion architecture, such as cross-attention [37] or embedded deep vision transformers (ViTs) [36], have introduced additional types of activations. Hence, as shown in upper Figure 1, only a small fraction of potential activations in modern diffusion models have been evaluated for their discriminative ability, which might hinder future work in this direction. For example, lower Figure 1 shows that Stable Diffusion XL (SDXL) [36], which is more advanced than Stable Diffusion v1.5 (SDv1.5) [37], fails to achieve better performance with the feature selection solution proposed in [2].

In this paper, we revisit the fundamental problem of feature selection, considering a more comprehensive candidate pool of activations. Due to its large volume, a full-scale quantitative comparison is no longer operational, urging us to modify the previous research methodology in [2]. As illustrated in Figure 1, instead of a direct quantitative comparison, we first explore the properties of diffusion U-Nets. These properties allow us to qualitatively and efficiently filter out many activations that are highly likely to be sub-optimal, shrinking the candidate pool for the quantitative comparison. More importantly, we find these properties are universal among diffusion models, making it possible to generalize our findings to more models beyond those covered in this paper.

Specifically, the properties we find, which are distinct to existing knowledge of model properties [1, 38], exactly correspond to three top-to-bottom levels of the diffusion U-Net: (i) **Diffusion noises** at the macro level: the diffusion process induces a new type of noise on both low- and high-frequency signals. (ii) **In-resolution granularity changes** at the in-resolution level: the changes in information granularity are not only across but also within resolutions. (iii) **Locality without positional embeddings** at the sub-module level: the embedded ViTs in diffusion U-Nets present a new type of local information different from that induced by the conventional positional embeddings [44]. Based on these insights, we develop effective feature selection solutions for several popular diffusion models. Finally, the experiments on three discriminative tasks, including semantic correspondence, semantic segmentation, and label-scarce segmentation, validate the superiority of our solutions over the SOTA methods.

In summary, the contributions of this work are three-fold:

- **Revisit of a fundamental problem**: To the best of our knowledge, we are the first to point out that the fundamental issue of feature selection remains unresolved in the realm of diffusion feature.
- **Generic insights**: The properties we find are universal among different diffusion U-Nets, which can provide valuable insights for future work.
- **Extensive validation**: Extensive experiments on three discriminative tasks validate the effectiveness of the feature selection solutions induced by our insights.

## 2 Related Work

**Diffusion models for discrimination.** We discuss three mainstream ways to use diffusion models for discrimination. (i) Diffusion classifier [26, 7] utilizes Bayes' theorem to transform a pre-trained diffusion model into an image classifier. This method enjoys a theoretical guarantee and does not need additional training. However, it is limited to image-level tasks. (ii) The second way is to model discriminative tasks as image-to-image generation tasks with diffusion models [22, 23]. This method is suitable for various dense vision tasks but requires heavy training. (iii) Diffusion feature [2, 58, 56, 41, 29], the focus of this paper, follows the traditional practice of feature extraction to pursue the balance between wider applicability and less training. This makes it adaptable for different tasks and alleviates the training needs for the diffusion model. Only small downstream models may require training.

The diffusion feature approach has seen various improvements. Some techniques toggle the input settings of diffusion models, such as seeking better timesteps [60, 29]. The others add trainable parameters outside the diffusion model to refine the outputs or provide an efficient fine-tuning alternative [58, 27, 46]. Additionally, some studies explore completely training-free methods that utilize spatial attention information [51, 43], and some attempts focus on text-free diffusion models [32, 33].

Despite the progress, previous methods only consider activations between neighboring modules, leaving many potential candidates unevaluated. Our work shows that such an overlook can hinder model performance. By introducing a more comprehensive feature selection solution, our method could generically enhance both existing and future diffusion feature approaches.

**Analysis on model properties.** The inner properties of neural networks have consistently received much attention [5, 34]. For transformers, Geva *et al.* [15] show that the feed-forward layers act as key-value memories and are interpretable for humans. Amir *et al.* [1] extend this insight to vision transformers and put it into practical applications such as image classification, while Vilas *et, al.* [45] try to make more detailed interpretation of ViT activations.

The methodology of these studies provides valuable guidance for our research. However, since diffusion models are trained for generation, our study relies more on qualitative analysis and feature visualization compared to previous work.

## 3 Preliminaries: Architecture of Diffusion U-Nets

Diffusion models typically involve a forward pass and a reverse pass [21]. During the forward pass, noise is gradually added to a clean image $x_0 \in \mathbb{R}^{3 \times w \times h}$ until the image resembles Gaussian noise, where $w$ and $h$ denote the width and height, respectively. This process can be denoted by $x_t \sim q(x_t|x_0)$, where $q$ represents the noise posterior and $t$ is the timestep. In the reverse pass, an end-to-end neural network $\epsilon_\theta$, as parameterized by $\theta$, learns to predict the noises and thus reconstruct the image. One such denoising step can be denoted as $\epsilon = \epsilon_\theta(x_t, t, c)$, where $\epsilon$ is the predicted noise, and $c$ is the condition that describes the expected image content. Although there are alternative formulations for diffusion models [39, 28, 40], they all rely on this neural network backbone $\epsilon_\theta$. This study focuses on this backbone, typically implemented as U-Net [38], rather than other components of diffusion models. Next, we will detail the architecture of the diffusion U-Net and standardize the *terminology* referring to different parts of the model, as shown in Figure 2.

We start with an overview. Following the initial U-Net design [38], the U-Net has three main *stages*: down-stage, mid-stage, and up-stage. The down-stage reduces the resolution of activations, while the up-stage increases it. Both stages contain multiple *resolutions*, in each of which the activations share

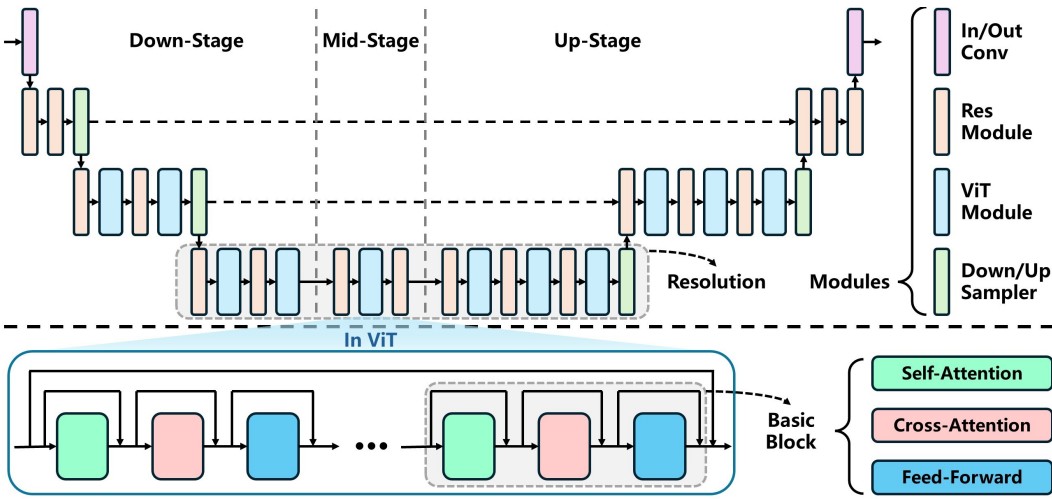

Figure 2: U-Net architecture (upper) and the ViT module (lower), taking SDXL as an example.

the same resolution. Furthermore, each resolution includes several *modules*, including ResModule (convolutional ResNet [20] structures), ViT Module [12], and Downsampler/Upsampler (simple convolutional layers). Previous diffusion feature approaches only consider activations between these adjacent modules, *i.e.*, inter-module activations.

We next dive into the details below the module level. Among these modules, ResModule and ViTs adopt *residual connection* [20], where an increment activation is added element-wise to the residual activation to refine it. Specifically, ResModule uses simple convolutional layers to produce increments, whereas ViTs use complex attention mechanisms, which will be further explained next. Typically, ViT operates as a standalone model followed by a decoder that produces the output predictions for visual tasks [12, 25, 9, 18]. However, in the diffusion U-Net, multiple ViTs are embedded into the network, and their outputs serve as increment activations.

Furthermore, each ViT module consists of several stacked *basic blocks*. A basic block typically has a self-attention layer to perform attention on the image itself and a feed-forward layer, essentially a two-layer MLP [15]. Modern diffusion U-Net introduces an additional cross-attention layer between the two layers, enabling the fusion of the image and additional textual prompts. Besides, each layer includes a residual connection, meaning that the increment activation added to the outer residual is also the internal residual.

In a nutshell, the architecture described above provides abundant activations that can serve as dense features. Given the massive activations, it is no longer operational to perform a full-scale quantitative comparison. Hence, we next make a comprehensive analysis of model properties to better understand these activations, which can help the qualitative filtering.

## 4 Distinct Properties of Diffusion U-Nets

The diffusion U-Net has many interesting properties, but now we only focus on those distinct from the knowledge of traditional U-Net [38] or ViTs [44]. As shown in Figure 3, this section highlights three noticeable properties, each of which corresponds to a different level of the diffusion U-Net architecture described in Section 3. Notably, **these properties can be universally observed in different samples and diffusion models, though all visualization in the main content is conducted on the same image and SDXL model for consistency**. Additional visualization is provided in Appendix A. Besides, the omitted properties are available in Appendix E.

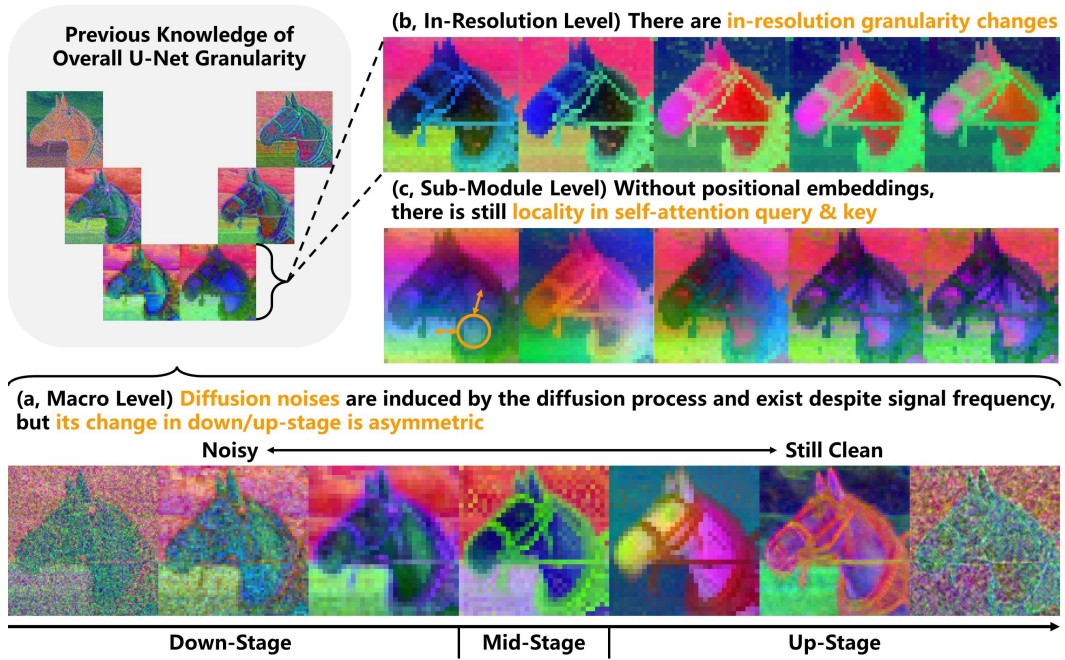

Figure 3: We highlight three properties of diffusion U-Nets that **are distinct from existing knowledge about other models**: (a) Asymmetric diffusion noises. (b) In-resolution granularity changes. (c) Locality without positional embeddings: pixels within the orange circle resemble nearby background pixels more than distant pixels on the horse's neck that are semantically closer.

## 4.1 Asymmetric Diffusion Noises

The first property is at the macro level and closely related to the overall diffusion process. It is common that high-frequency signals are typically noisy [5, 34]. This phenomenon can also be observed in diffusion U-Nets, especially within the increment activations of residual connection. However, this does not mean that low-frequency signal activations in diffusion U-Nets are free from noises. As shown in Figure 3(a), the diffusion process introduces a new type of noise that also impacts low-frequency signals. This is not surprising since the diffusion process requires the backbone to process noisy inputs and predict noises as outputs. As a result, activations near the inputs or outputs, regardless of their frequency, also suffer from such noises. Considering the special cause, we name such noises *diffusion noises*.

How does the influence of diffusion noises spread across diffusion U-Nets? As shown in Figure 3(a), diffusion noises exist throughout the entire down-stage, with a decreasing magnitude. Remarkably, during the early half of the up-stage, activations are rather clean, with no perceivable diffusion noises. Only in the later half do diffusion noises start to resurface. The existence of this asymmetric behavior can also be indirectly supported by the ablation curves in many fellow studies such as [2, 32, 33]. Furthermore, as shown in Appendix A, such an asymmetric pattern is consistent across activations in different diffusion models.

Similar to common high-frequency noises, diffusion noises can degenerate feature quality. Hence, this property can serve as a criterion for identifying and filtering out sub-optimal activations, which we will delve into in Section 5.

## 4.2 In-Resolution Granularity Changes

The second property is at the in-resolution level and closely related to recent advances in diffusion architectures. Specifically, the design of U-Net follows the idea of resolution hierarchy [38]. Consequently, the overall architecture displays a fine-coarse-fine granularity trend, looking like the alphabet "U". Traditional U-Nets implement this architecture with a relatively large number of resolutions, while each resolution is typically small, equipped with two or three simple convolutional layers.

Hence, the understanding of traditional U-Net focuses on the granularity changes across resolutions, implicitly assuming that the change within a single resolution is negligible.

However, diffusion U-Nets become much "fatter". In other words, modern diffusion U-Nets typically have much fewer resolutions, but each resolution is significantly enlarged. For example, SDv1.5 only has four resolutions [37], and SDXL further decreases this number to three [36], as shown in Figure 2. Meanwhile, each resolution can produce much more activations, primarily thanks to the embedded ViT modules. This architectural evolution makes the granularity change within a single resolution more significant, as depicted in Figure 3(b).

Different granularity carries varied information and quality, resulting in different discriminative performance on downstream tasks [2, 29]. Hence, this discovery of *in-resolution granularity changes* highlights the necessity to evaluate more activations, especially those within the embedded ViTs, as they are of different granularity.

### 4.3 Locality without Positional Embeddings

For the third distinct property, we delve into the sub-module level, *i.e.*, the blocks in embedded ViT modules. Positional embeddings, which are widely used in language transformers [44] and ViTs [12], aim to provide spatial information for each input token. Consequently, the activations of traditional ViTs display strong positional information, where the latent pixel resembles nearby pixels more than those that are semantically similar but far away [1]. This phenomenon is significant for the layers close to the inputs. When the layer goes deeper, the tokens are refined with semantic information, making the activations display less positional information. However, only in the last few layers, such positional information becomes negligible.

In contrast, ViT modules in diffusion U-Nets do not use positional embeddings [37, 36], perhaps because the other U-Net components have provided sufficient spatial cues. This change results in distinct properties of the activations. On one hand, positional information is negligible for most activations despite how near they are to the inputs. For example, even in the first basic block, cross-attention query activations contain no perceivable positional information. On the other hand, the queries and keys of self-attention still display non-negligible positional information, marked with orange circles in Figure 3(c). Specifically, the latent pixel on the horse's neck is a light blue color, similar to the pixels to its left that actually represent the background. In contrast, the pixels above the circle are in purple color, though they also represent the horse's neck. Such comparison shows that a latent pixel is more similar to other pixels that are spatially near it than those semantically closer to it.

We name this phenomenon *locality* since it has a different mechanism from that induced by positional embeddings. As pointed out in [12], self-attention allows ViT to integrate global and local information even in the shallow layers, and the attention scope enlarges *w.r.t.* layer depth. Even without positional embeddings, self-attention activations are generally consistent with this insight, leading to the existence of locality. Nevertheless, the magnitude has indeed greatly reduced, compared to the visualization of conventional ViT activations in [1]. As shown in Figure 3(c), in shallow activations, locality exists but is inherently weaker, as much semantic information is still reserved. In addition, locality quickly diminishes as the layer goes deeper.

Since positional information can degrade the quality of activations [1], its absence has the potential to enhance the activations in ViT modules. Moreover, locality can play a special role in activation filtering, as presented in Section 5.

### 4.4 Universality of Three Properties

Although the visualization in Figure 3 is conducted only on SDXL, the scope of these properties is not limited to the specific architecture. Further supporting evidence is available in Appendix A.

(i) Diffusion noises directly arise from the diffusion process. Hence, it is promising to extend this property to other diffusion backbones.

(ii) In-resolution granularity changes come from the "fatness" of U-Nets, making it potentially applicable to more traditional U-Nets.

(iii) Locality originates from the self-attention mechanism in ViT architectures, so it is broadly applicable to standalone or embedded ViTs where positional embeddings are absent.

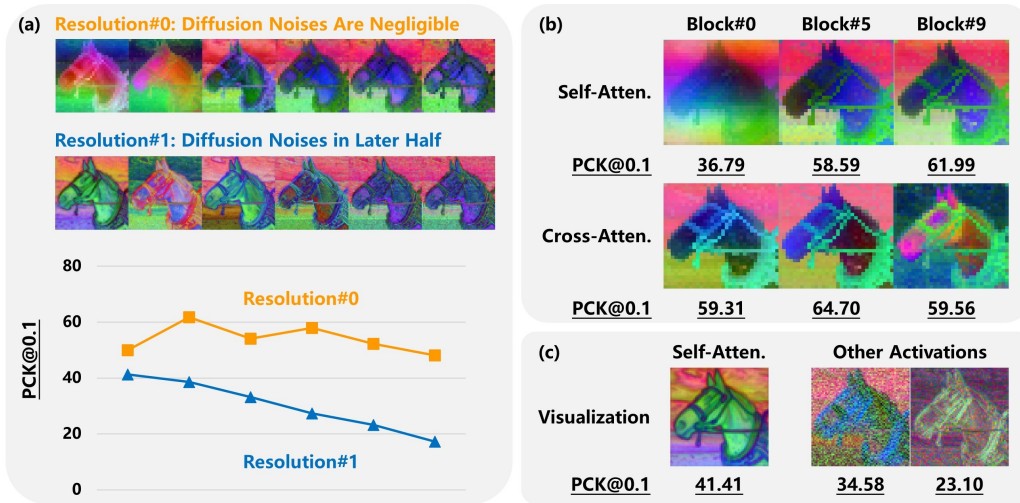

Figure 4: (a) Diffusion noises result in a significant performance degeneration (Resolution#1). (b) Locality degrades the quality of self-attention activations (Block#0 and Block#5). (c) Locality in self-attention activations can suppress diffusion noises, leading to better quality than noisy activations (41.41 *v.s.* 34.58). All PCK@0.1$_{\text{img}}$($\uparrow$) results are evaluated on the semantic correspondence task.

## 5 Enhanced Feature Selection from Diffusion U-Nets

So far, we have had a more comprehensive understating of the properties of diffusion U-Nets. All these properties, especially in-resolution granularity changes, encourage us to reconsider the feature selection solution, with a special emphasis on the activations in ViT modules. With these properties, we are also able to filter out many low-quality activations qualitatively, followed by a thus simplified quantitative comparison.

### 5.1 Qualitative Filtering

**Avoiding Diffusion Noises.** As shown in Figure 4(a), diffusion noises tend to degrade the quality of activations. Hence, it is natural to filter out the activations severely affected by diffusion noises from the candidate pool. Specifically, according to the asymmetric trend of diffusion noises, **we only consider activations in the early half of the up-stage**, which are rather clean. This approach will significantly reduce the number of candidate activations and simplify the quantitative comparison.

**Avoiding Self-Attention Locality.** The locality in self-attention modules is another important factor that can degrade the activations. The empirical evidence in Figure 4(b) demonstrates that these activations are generally inferior to the others, such as those from cross-attention layers or the outputs of ViT basic blocks. Consequently, it is rather safe to **filter out most activations in self-attention modules** from our candidate pool.

**Using Locality to Suppress Diffusion Noises.** So far, the activations in the candidate pool are clean and free from locality. However, all these activations are low-resolution ones since high-resolution activations are generally noisy and thus filtered out. This is unfavorable since some detailed information might only exist in high-resolution activations. To address this issue, we exploit a side effect of self-attention locality. Specifically, as indicated in Figure 4(c), locality can help suppress diffusion noises via its focus on spatial structures. Although locality is sub-optimal, it is still superior to severe diffusion noises. In view of this, the candidate pool **reserves self-attention activations extracted from the later half of the up-stage**.

We have filtered out many candidate activations based on the distinct properties of diffusion U-Nets. Additionally, all increment activations in residual connection can be further filtered out since they introduce high-frequency noises [5, 34]. After such qualitative filtering, a small but high-quality candidate pool is available. Taking SDXL as an example, the number of candidates decreases from 279 to 63, *i.e.*, a 78% reduction. Next, we explain how to conduct this quantitative comparison briefly.

Table 1: Experimental results on the semantic correspondence task. The best results are in **bold** font and the runner-up is underlined.

| Category | Method | PCK@$0.1_{\text{img}}$ ↑ | PCK@$0.1_{\text{bbox}}$ ↑ |
|---|---|---|---|
| SOTA | DINO | 51.68 | 41.04 |
| | DHPF | 55.28 | 42.63 |
| | DIFT | - | 52.90 |
| | DHF | 72.56 | 64.61 |
| Baseline | Legacy-v1.5 | 75.14 | 66.73 |
| | Legacy-XL | 66.00 | 59.16 |
| Ours | Ours-v1.5 | 77.78 | 69.83 |
| | Ours-XL | 81.72 | 75.18 |
| | Ours-XL-t | **83.90** | **76.86** |

## 5.2 Quantitative Comparison

The quantitative comparison follows the protocol of [2]. Specifically, given an input image $x_0$, we first perform the forward pass with a pre-defined timestep $t$ to generate the noisy image $x_t$. Then, the U-Net backbone conducts one denoising step. Instead of collecting the model output $\epsilon$, we gather U-Net activations and consolidate them to the candidate pool, as described in Section 5.1. Afterward, each activation is individually fed to a downstream model to evaluate its discriminative potential, with the details of the model described in Appendix C. Such comparison is made fair by using the same dataset for all activations. Finally, we can obtain the capability ranking of activations from each resolution, according to which features can be selected wisely. Notably, we conduct such comparisons on multiple datasets to guarantee generalizability, since the best activations may differ among different scenes [2]. Thanks to qualitative filtering, **the time cost for each (model, dataset) pair has been reduced by more than one week**, equipped with Nvidia(R) RTX 3090 GPUs.

Given the capability ranking, feature selection is in fact flexible, as it is possible to combine multiple activations and specifically tune the choice for a task. For practicality, we provide off-the-shelf feature combinations for SDv1.5 and SDXL that are likely to generically perform well in Appendix B, according to the results detailed in Appendix D. **Each of them consists of four activations mainly from ViT modules rather than inter-module positions**. Taking SDv1.5 as an example, one of the four selected features is from one self-attention layer in the highest resolution, which utilizes our observation in Section 4.

## 6 Experimental Validation on Multiple Discriminative Tasks

To validate the effectiveness of our feature selection solution, the experiments are conducted on three popular discriminative tasks: semantic correspondence, semantic segmentation, and label-scarce segmentation. The SOTA methods for each task are selected as competitors. Besides, we also compare with two baselines that select the conventional inter-module activations as features [2], named *Legacy-v1.5* and *Legacy-XL*. For our method, we provide the following implementations:

- *Ours-v1.5 & Ours-XL*: Features extracted from SDv1.5 and SDXL, respectively.
- *Ours-XL-t*: For fairness, we further enhance SDXL features with some additional techniques that are also adopted by SOTAs. The techniques we select, *i.e.*, attention maps [58] and feature amalgamation [2, 29, 60], are relatively simple and lightweight.

More experimental details are in Appendix C, such as task information, evaluation metrics, and implementation details. Besides, Appendix D presents additional results not covered here.

### 6.1 Empirical Results on Semantic Correspondence

We present the experimental results for semantic correspondence in Table 1. From the results, we have the following observations:

Table 2: Experimental results on semantic segmentation and its altered version with scarce labeled data, evaluated using mIoU↑ metric. The best results are in **bold** font and the runner-up is underlined.

| Category | Method | Standard Setting | | Method | Label-Scarce Setting |
| | | ADE20K | CityScapes | | Horse-21 |
|---|---|---|---|---|---|
| SOTA | MaskCLIP | 23.70 | - | SwAVw2 | 54.0 ± 0.9 |
| | ODISE | 29.90 | - | MAE | 63.4 ± 1.4 |
| | VPD | 37.63 | 55.06 | DatasetDDPM | 60.8 ± 1.0 |
| | Meta Prompts | 40.89 | 71.94 | DDPM | 65.0 ± 0.8 |
| Baseline | Legacy-v1.5 | 40.26 | 64.01 | Legacy-v1.5 | 59.4 ± 1.3 |
| | Legacy-XL | 27.78 | 71.67 | Legacy-XL | 53.0 ± 0.9 |
| Ours | Ours-v1.5 | 41.07 | 64.10 | Ours-v1.5 | 60.2 ± 0.9 |
| | Ours-XL | 43.45 | 74.47 | Ours-XL | 62.7 ± 0.7 |
| | Ours-XL-t | **45.71** | **75.89** | Ours-XL-t | **66.3 ± 0.9** |

(i) *Ours-v1.5* outperforms *Legacy-v1.5* (77.78 *v.s.* 75.14 on PCK@$0.1_{img}$). The main reason is that *Legacy-v1.5* fails to effectively handle the diffusion noises in high-resolution activations. Previous approaches either reserve the noisy activations and thus suffer from performance degradation [2, 52], or simply discard high-resolution activations and thus suffer from information loss [58, 46]. In contrast, our approach uses self-attention locality to suppress diffusion noises and harvest better high-resolution activations.

(ii) *Legacy-XL* is inferior to *Legacy-v1.5* (66.00 *v.s.* 75.14 on PCK@$0.1_{img}$). At first glance, this result is counter-intuitive since SDXL is more advanced than SDv1.5. However, the analysis in Section 4 can unravel the mystery. Specifically, since SDXL has more ViT modules, the more valuable activations shift from inter-module positions to these embedded ViTs. Since the baseline does not consider ViT modules, *Legacy-XL* fails to achieve better performance. In contrast, *Ours-XL* shows improvement over *Ours-v1.5* (81.72 *v.s.* 77.78 on PCK@$0.1_{img}$). This is consistent with the advance in model architecture and again validates our analysis.

(iii) *Ours-XL-t* significantly outperforms the SOTA method with a similar amalgamation technique, *i.e.*, DHF [29] (**83.90** *v.s.* 72.56 on PCK@$0.1_{img}$). This performance gain again validates the effectiveness of our method.

## 6.2 Empirical Results on Semantic Segmentation

As shown in the left part of Table 2, *Ours-XL-t* and *Ours-XL* achieve state-of-the-art performance on the semantic segmentation task (**45.71** and 43.45 *v.s.* 40.89 on ADE20K), demonstrating its effectiveness and generalizability. Furthermore, the most competitive SOTA, Meta Prompts [46], introduces a large number of trainable parameters and uses the diffusion U-Net recurrently, which is rather time-consuming. In contrast, our method delivers superior results with efficiency maintained.

Unlike the results in semantic correspondence, *Legacy-XL* outperforms *Legacy-v1.5* on CityScapes (71.67 *v.s.* 64.01), and the performance gap between *Legacy-v1.5* and *Ours-v1.5* is narrow (64.01 *v.s.* 64.10). This is because this task utilizes a relatively large-scale downstream model, which can significantly refine the input features and thus reduce the gap in feature quality. Nevertheless, *Ours-XL* still achieves a significant improvement over *Legacy-XL* (74.47 *v.s.* 71.67).

## 6.3 Empirical Results on Label-Scarce Segmentation

One advantage of diffusion features is the applicability to label-scarce scenarios [2]. For validation under such conditions, we experiment on the label-scarce segmentation task, with results presented in the right part of Table 2. The observations are generally similar to those on the semantic correspondence task. For example, *Ours-v1.5* outperforms *Legacy-v1.5* (60.2 *v.s.* 59.4), *Legacy-XL* is inferior to *Legacy-v1.5* (53.0 *v.s.* 59.4), and *Ours-XL* is better than *Ours-v1.5* (62.7 *v.s.* 60.2).

Next, we focus on the comparison with the SOTA method, DDPM [2]. Although DDPM is a relatively early study, it outperforms the other competitors and our most implementations. This result has two reasons. On one hand, DDPM performs the diffusion process directly in the image space

rather than the currently common practice of compressed latent space. Although inefficient, such an implementation yields better discriminative features. On the other hand, DDPM utilizes diffusion models specifically trained on each dataset. In comparison, we use pre-trained general-purposed SDv1.5 or SDXL to be more efficient and thus consistent with the motivation of diffusion feature. Hence, it is rather challenging to surpass this SOTA method. Fortunately, our best implementation, *Ours-XL-t*, achieves this goal with the help of additional lightweight techniques (**66.3** *v.s.* 65.0).

## 7 Conclusion and Future Work

In this study, we revisit the fundamental problem of feature selection from diffusion U-Nets. We point out that prior arts only consider a limited range of potential activations. In contrast, we consider a much wider range of activations as candidates, especially those extracted from the embedded ViT modules. Given the large volume of the candidate pool, we first analyze the properties of diffusion U-Nets. The properties we find are universal such that our observations are not limited to the specific diffusion architecture. Based on these properties, we qualitatively filter out many activations with low quality, facilitating the following quantitative comparison. On top of this, concrete feature selection solutions are proposed for two popular diffusion models, *i.e.*, SDv1.5 and SDXL. Finally, extensive experiments on three discriminative tasks validate the effectiveness of our method.

However, we are not sure whether our observations can generalize well to recently-developed DiT models [35] since they have a markedly different architecture from U-Net-based diffusion models. Thus, analyzing DiT models is a promising topic for future research.

## Acknowledgments

This work was supported in part by the National Key R&D Program of China under Grant 2018AAA0102000, in part by National Natural Science Foundation of China: 62236008, U21B2038, U23B2051, 61931008, 62122075 and 62025604, in part by Youth Innovation Promotion Association CAS, in part by the Strategic Priority Research Program of the Chinese Academy of Sciences, Grant No. XDB0680000, in part by the Innovation Funding of ICT, CAS under Grant No.E000000, in part by the China National Postdoctoral Program for Innovative Talents under Grant BX20240384.

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

# Contents

# A    Additional Visualization for Distinct Properties of Diffusion U-Nets

## A.1    Visualization of Other Sample Images

We have provided activation visualization from SDXL in a simple outdoor scene presenting a horse in the wild. Next, we will show visualization from the same model in different scenes to demonstrate the universality of the observed properties. Another simple outdoor scene presenting a cat is in Figure 5. Two simple indoor scenes are visualized in Figure 6 and Figure 7. Two complex outdoor scenes of urban streets are visualized in Figure 8 and Figure 9. Two complex indoor scenes are shown in Figure 10 and Figure 11.

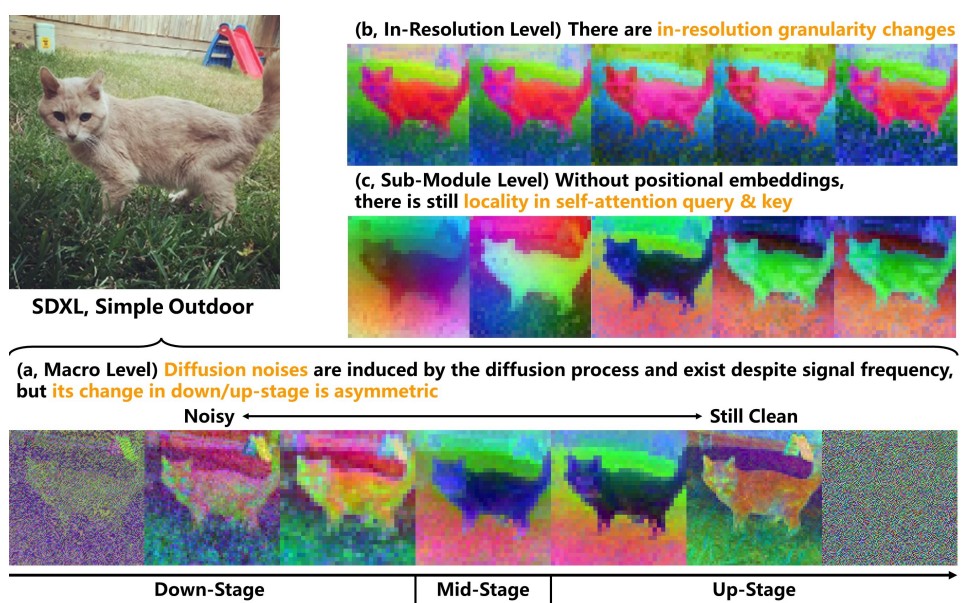

Figure 5: Visualization of SDXL activations on a simple outdoor scene.

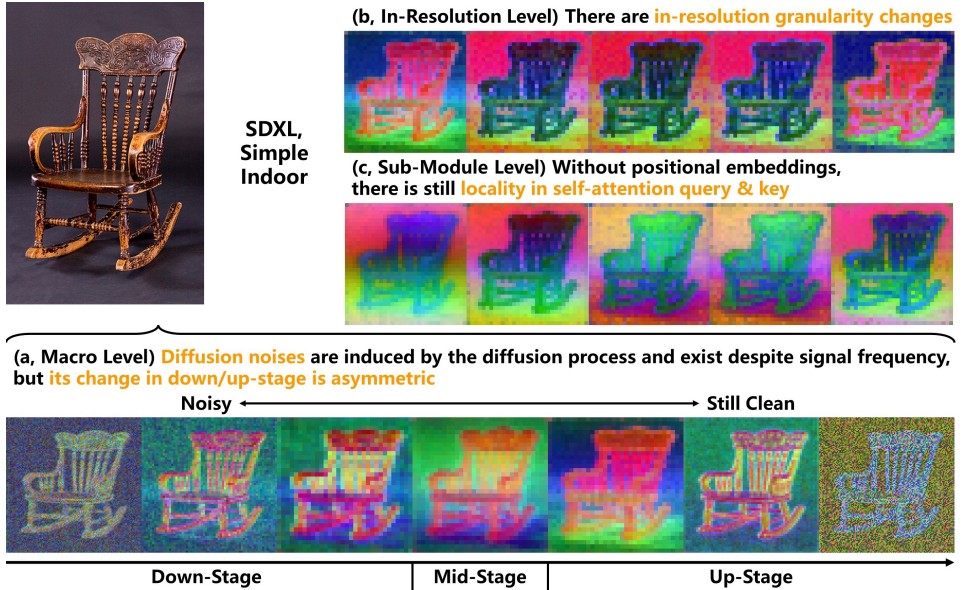

Figure 6: Visualization of SDXL activations on a simple indoor scene.

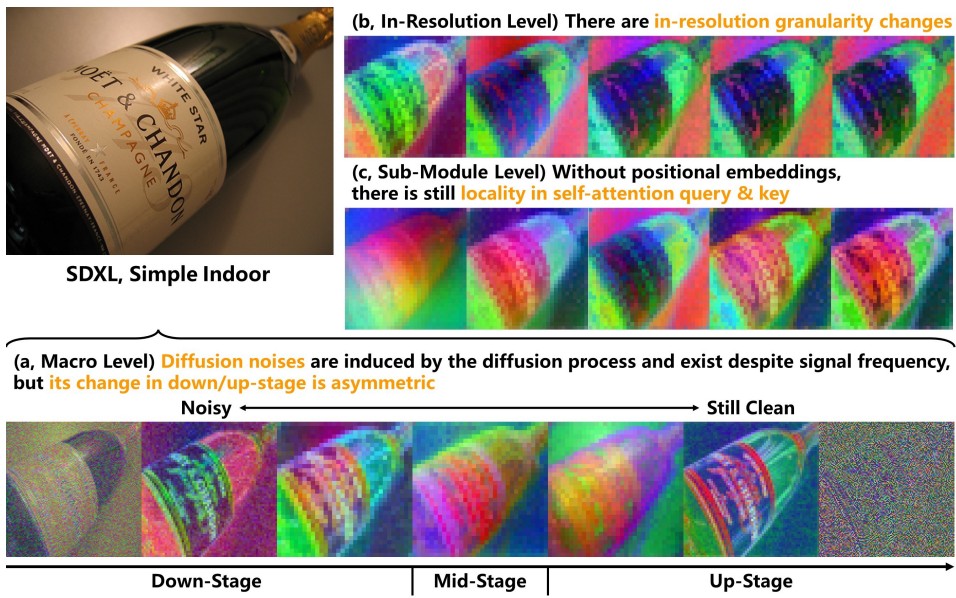

Figure 7: Visualization of SDXL activations on a simple indoor scene.

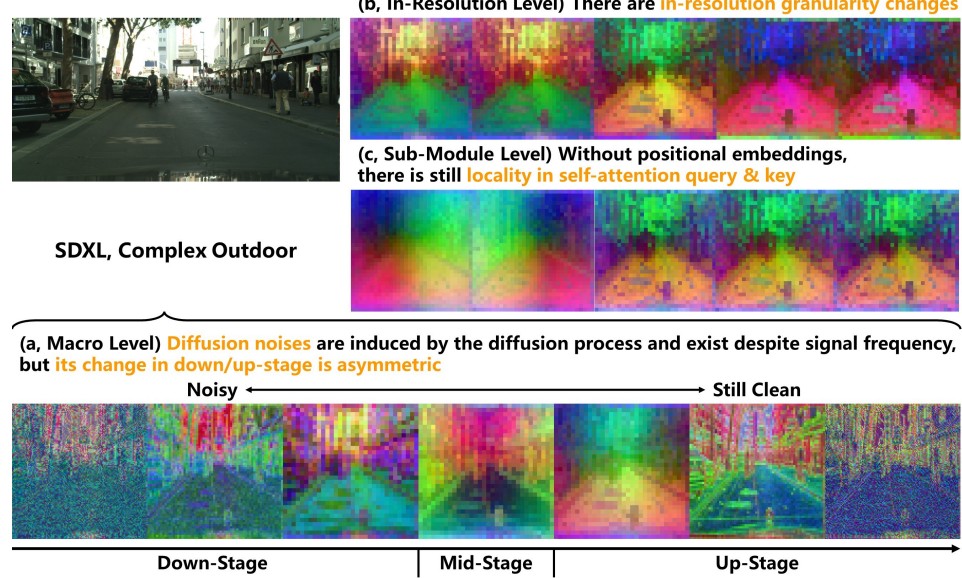

Figure 8: Visualization of SDXL activations on a complex outdoor scene.

## A.2 Visualization from Other Diffusion Models

**Playground v2 Activations.** Playground v2[2] shares the same architecture as SDXL but is trained independently, and it is claimed to be more powerful in generation. In Figure 12, we present its activation visualization using the same horse image as the primary SDXL visualization. Compared to SDXL, Playground v2 activations are less noisy, particularly in the down-stage. This supports the claim that Playground v2 is a stronger model.

**SDv1.5 Activations.** SDv1.5, as an older model, has a slightly different architecture from SDXL. Specifically, SDv1.5 has four resolutions instead of three, but its ViTs contain only one layer. Moreover, in SDXL, only the highest resolution lacks ViT due to efficiency concerns, while in

---

[2] https://huggingface.co/playgroundai/playground-v2-1024px-aesthetic

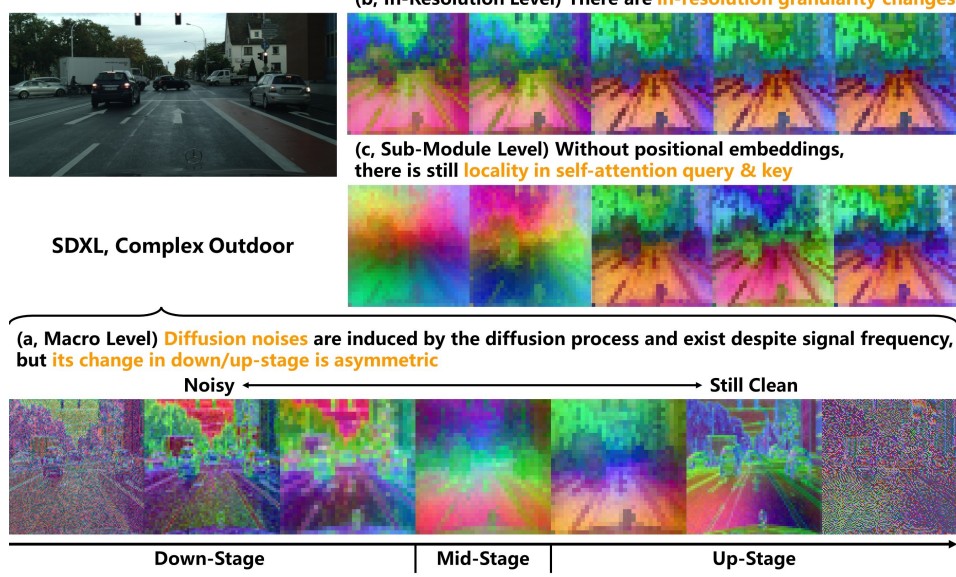

Figure 9: Visualization of SDXL activations on a complex outdoor scene.

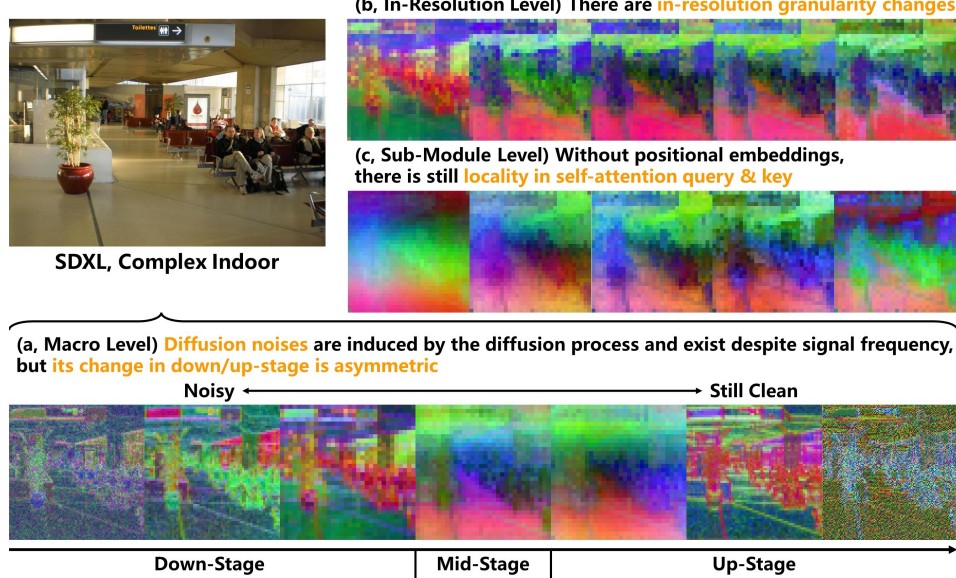

Figure 10: Visualization of SDXL activations on a complex indoor scene.

SDv1.5, only the lowest resolution lacks ViT due to its low resolution. Despite the architectural changes, the three unique properties still apply to SDv1.5, as shown in Figure 13.

**Video Diffusion Activations.** To further demonstrate the universality, we even select a diffusion model for video generation [6] for visualization. This model is based on the SDv1.5 architecture, with an additional temporal attention layer inserted after cross-attention to enable sequential generation. Although this model is designed for a different task, we can still observe the three unique properties in Figure 14.

**Conventional U-Net Activations.** As comparison to diffusion U-Net, we also visualize some activations from a conventional U-Net in Figure 15, where the three properties of diffusion U-Net can hardly be observed.

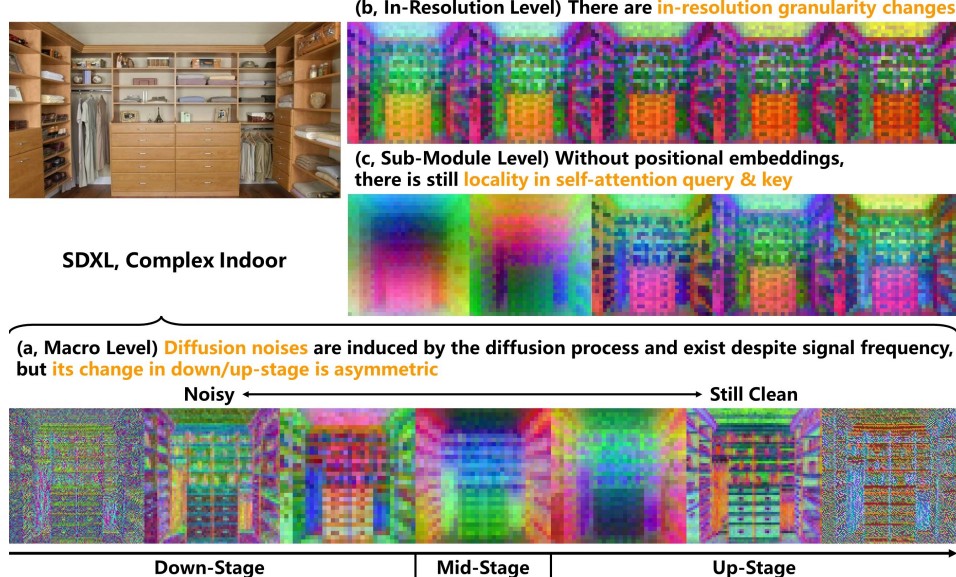

Figure 11: Visualization of SDXL activations on a complex indoor scene.

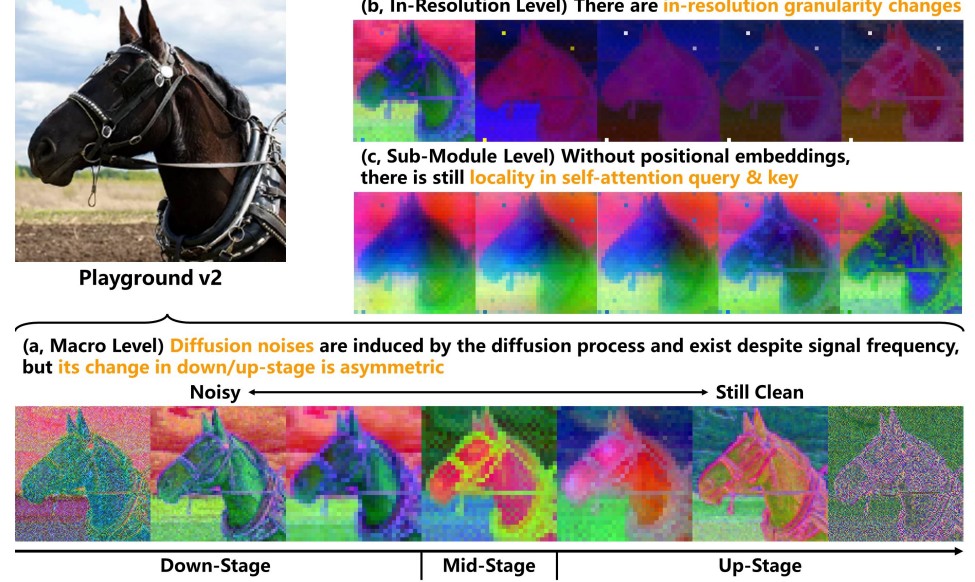

Figure 12: Visualization of Playground v2 activations on a simple outdoor scene.

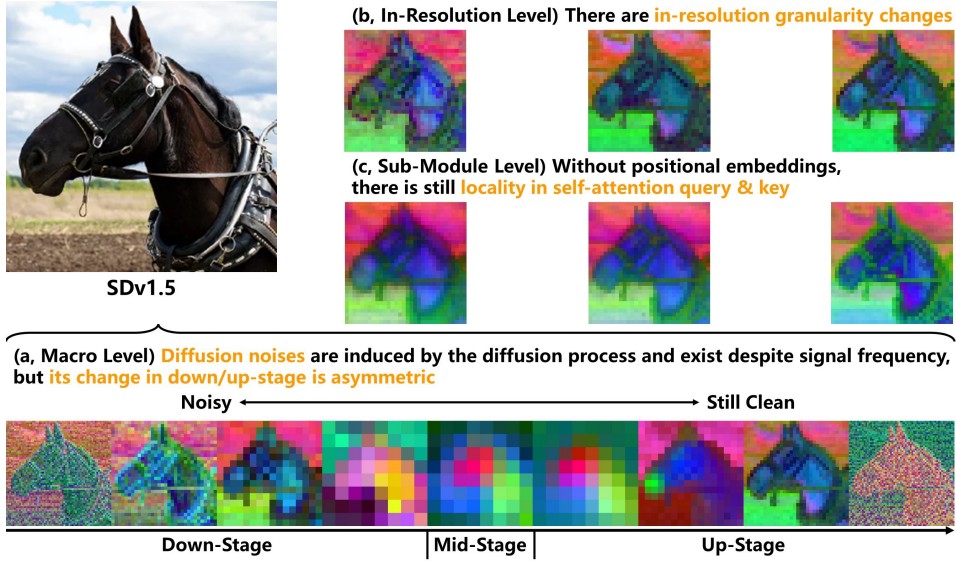

Figure 13: Visualization of SDv1.5 activations on a simple outdoor scene.

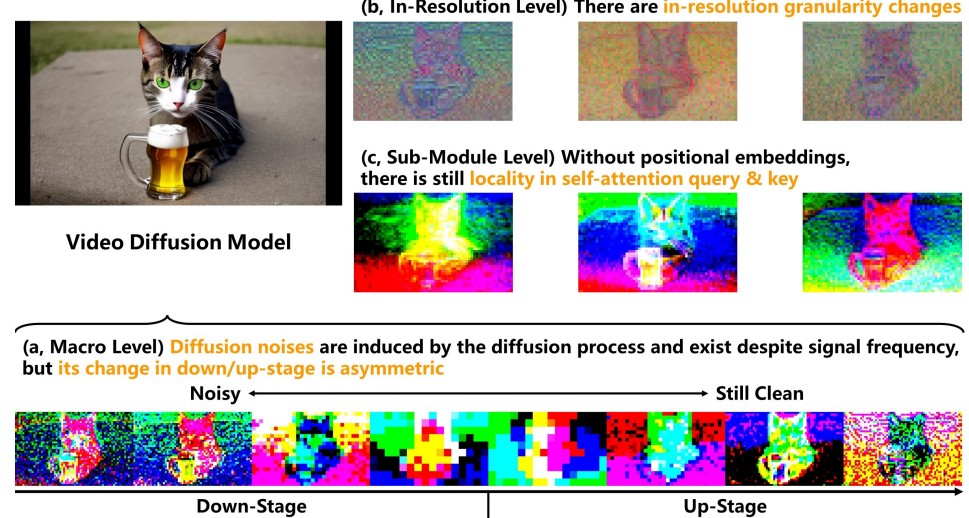

Figure 14: Visualization of the activations of a video diffusion model. The input is a short video, and we visualize one frame of it.

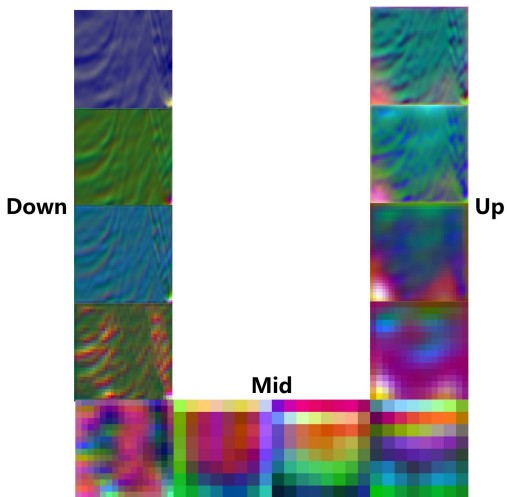

Figure 15: Activation visualization from a traditional U-Net for semantic segmentation [57].

# B   Details of Our Method

In this section, we index all diffusion U-Net components in the order in which they are activated during a network forward run. Besides descriptions in natural language, we also index the activations using the same notations as used in our code implementation. These notations denote stages using *down/mid/up*, resolutions as *level*, and module index as *repeat*. Index starts to count from 0 instead of 1, following the convention in coding.

## B.1   Feature Selection Solution for SDv1.5

We select four activations from SDv1.5, maintaining the same total feature channels as the conventional way to extract all output activations from each resolution.

   (i) The cross-attention query activation from the 2nd ViT, the 2nd resolution. This provides coarse information for simple scenes (up-level1-repeat1-vit-block0-cross-q).

  (ii) The inter-module activation after the 3rd ResModule, the 2nd resolution. This provides coarse information for complex scenes (up-level1-repeat2-res-out).

 (iii) The cross-attention query activation from the 2nd ViT, the 3rd resolution. This provides finer information with a higher resolution (up-level2-repeat1-vit-block0-cross-q).

 (iv) The self-attention key activation from the 1st ViT, the 4th resolution. This extracts features from the highest resolution, harnessing the noise suppression effect of self-attention locality (up-level3-repeat0-vit-block0-self-k).

We omit the index of basic blocks in ViTs, as SDv1.5 only contains one-layer ViTs. Additionally, we totally ignore the lowest resolution, as its activations have very low resolution ($8 \times 8$ if the input image is $512 \times 512$), suggesting inferiority, as supported by the quantitative comparison.

## B.2   Feature Selection Solution for SDXL

Four activations are selected from SDXL, trying to get similar total feature channels to the SDv1.5 feature selection solution.

   (i) The output activation after the 8th basic block, the 1st ViT, the 1st resolution. This provides relatively coarse information for simple scenes (up-level0-repeat0-vit-block7-out).

  (ii) The output activation after the 6th basic block, the 1st ViT, the 1st resolution. This provides relatively coarse information for complex scenes (up-level0-repeat0-vit-block5-out).

 (iii) The cross-attention query activation from the 1st basic block, the 1st ViT, the 2nd resolution. This provides relatively fine information for simple scenes (up-level1-repeat0-vit-block0-cross-q).

 (iv) The output activation after the 1st basic block, the 1st ViT, the 2nd resolution. This provides relatively fine information for complex scenes (up-level1-repeat0-vit-block0-out).

We ignore the highest resolution since it is affected by diffusion noises and lacks ViTs from which we can extract self-attention activations. However, if the downstream model is strong enough to learn to suppress noises, it may be possible to additionally extract inter-module activations from this resolution to harness more information.

## B.3   Feature Selection Solution with Additional Techniques

For the setting *Ours-XL-t*, we mainly utilize two simple techniques that are also adopted in some SOTA methods: (i) We additionally extract attention maps, *i.e.*, the similarity scores of cross-attention query and key, as dense features [58, 52]. Such attention maps are closely related to the semantics of prompts, thus providing important supplementary information. Despite its usefulness, this technique only adds a few additional channels to the features. (ii) We amalgamate features from different models [2, 29, 60] through simple concatenation. This technique is a common practice and can be seen as an extension of amalgamating different activations together as a whole feature. We next explain what activations are selected to implement the two techniquess.

We first extract features according to the feature selection solution for SDXL. Afterward, we extract additional features from SDv1.5:

(i) The cross-attention query activation from the 2nd ViT, the 2nd resolution (up-level1-repeat1-vit-block0-cross-q).

(ii) The cross-attention query activation from the 2nd ViT, the 3rd resolution (up-level2-repeat1-vit-block0-cross-q).

(iii) The upsampler output activation from the 3rd resolution, to harness high-resolution information (up-level2-upsampler-out).

(iv) The self-attention key activation from the 1st ViT, the 4th resolution, to harness high-resolution information (up-level3-repeat0-vit-block0-self-k).

(v) The attention maps averaged over all cross-attention layers in the up-stage.

We also extract one feature from Playground v2: the output activation after the 4th basic block, the 1st ViT, the 1st resolution (up-level0-repeat0-vit-block3-out).

### B.4 Alternative Feature Selection Solution with Additional Techniques

Large-scale datasets for semantic segmentation mostly consist of images of complex scenes. In such cases, we find attention maps can be too noisy to be useful. Therefore, we discard the attention map technique and the entire SDv1.5 model, as it is weaker compared to the newer SDXL and Playground v2 models. To compensate for the loss of activations, we select additional activations from SDXL and Playground v2.

From SDXL, we select all the activations as described in the feature selection solution for SDXL and select one additional activation: the upsampler output activation from the 2nd resolution, to harness high-resolution information (up-level1-upsampler-out). From Playground v2, we select the following activations:

(i) The output activation after the 6th basic block, the 1st ViT, the 1st resolution (up-level0-repeat0-vit-block5-out).

(ii) The cross-attention query activation from the 1st basic block, the 1st ViT, the 2nd resolution (up-level1-repeat0-vit-block0-cross-q).

(iii) The upsampler output activation from the 2nd resolution (up-level1-upsampler-out).

## C    Experimental Details

### C.1    Semantic Correspondence

**Task and Dataset.** Semantic correspondence [18] involves finding a pixel in an image that semantically matches another keypoint pixel in a reference image, such as the hind legs of two different cats. We conduct experiments on the SPair-71k dataset [30].

**Evaluation Metric.** PCK@$0.1_{img}$($\uparrow$) and PCK@$0.1_{bbox}$($\uparrow$) are used, following the widely-adopted protocol reported in [30]. These two metrics mean the percentage of correctly predicted keypoints, where a predicted keypoint is considered to be correct if it lies within the neighborhood of the corresponding annotation with a radius of $0.1 \times max(h, w)$. For PCK@$0.1_{img}$/PCK@$0.1_{bbox}$, $h, w$ denote the dimension of the entire image/object bounding box, respectively.

**SOTA Competitors.** We provide the results from four SOTA methods: DINO [4] and DHPF [31] as non-diffusion-feature methods, as well as DIFT [41] and DHF [29] as diffusion feature methods.

**Implementation Details.** The semantic correspondence task can be done via the nearest neighbor algorithm [42], which is unsupervised and training-free [30]. We add one additional trainable convolutional layer before applying the nearest neighbor algorithm to refine the input features, which is also adopted by some SOTAs including DHF. The model is trained for two epochs, each containing 5,000 sample pairs, following conventional settings. Our implementation is derived from DHF, and we keep all hyper-parameters at their default settings.

## C.2 Semantic Segmentation

**Task and Dataset.** Semantic segmentation [18] is essentially pixel-level classification. For this task, we choose the ADE20K dataset [59] with over 20k annotated images of 150 semantic categories, and the CityScapes dataset [8], which contains 5,000 fine annotated images of urban street scenes.

**Evaluation Metric.** We use mIoU metric, which is the mean over the IoU performance across all semantic classes [18]. For each image, IoU (Intersection over Union, ↑) is defined by #(overlapped pixels between the prediction and the ground truth) / #(union pixels of them).

**SOTA Competitors.** We choose three diffusion feature SOTA methods as competitors. ODISE [52] is an early method with a simple implementation. VPD [58] is another early study of this field, which introduces additional text adapter modules for improvement. Meta Prompts [46] is a newer method and shows significant improvements. We also report the performance of MaskCLIP [11], which is included as a competitor in the ODISE study.

**Implementation Details.** Both VPD and Meta Prompts perform full-scale fine-tuning on the diffusion U-Net using feedback from the discriminative task. This heavy fine-tuning does not entirely comply with the motivation of diffusion feature, which seeks a balance between wider applicability and less training, and is hard to extend to the larger SDXL model. Therefore, we keep the entire diffusion model frozen instead. As the setting has been changed, the performance of VPD and Meta Prompts in Table 2 is based on our experiments, not the reported results from the original papers. Our implementation directly uses the hyper-parameters reported in Meta Prompts.

## C.3 Label-Scarce Segmentation

**Task and Dataset.** Using features from a pre-trained diffusion model ensures good performance even when labeled training data is scarce [2]. For this setting, we use a dataset collected in [2] and experiment on its Horse-21 subset, the data of which is sourced from LSUN [54]. This subset contains only 30 labeled training images to be consistent with the intuition. The semantic segmentation in the label-scarce scenario also uses the mIoU metric.

**SOTA Competitors.** We select the SOTA diffusion feature approach, DDPM [2], as the major competitor. We also include other representative segmentation methods: DatasetDDPM, MAE [19], SwAV [3], which are all reported in [2].

**Implementation Details.** Following DDPM [2], the downstream model is an ensemble of ten simple MLP networks, each conducting pixel-wise classification. The simplicity of the model is intended to demonstrate the innate capability and generalizability of diffusion models. Our implementation is derived from DDPM with only batch size changed among all hyper-parameters. We use a larger batch size for faster experiments as a smaller one does not improve performance. **Additionally, this is also the setting for the quantitative comparison, as the compact size of the dataset can enhance efficiency**.

# D  Additional Experimental Results

## D.1 Generalizability across Different Scenes

As stated in Section 5, it is preferable to conduct the quantitative comparison across multiple datasets and choose activations that are optimal for each. This approach can enhance the generalizability of the selected features. To evaluate the generalizability of our features, we conducted an additional experiment, with results presented in Table 3.

In this experiment, we design an alternative feature selection solution for SDXL, based solely on quantitative results from a single dataset consisting of simple scenes. In this solution, we extract both optimal and slightly sub-optimal activations, maintaining the same total number of feature channels as the standard solution. The alternative solution achieves higher performance on the simple scene it is based on but performs significantly worse on the other scene. Therefore, we conclude that our standard feature selection solution achieves generalizability across different scenes, albeit with a slight performance drop compared to features specifically selected for each scene.

Table 3: Examination of generalizability across different scenes. *Generic Solution* refers to our standard feature selection solution for SDXL, while *Specific Solution* refers to the outcome of considering only a simple scene for the quantitative comparison. The experiment is conducted on the label-scarce segmentation task, where the Horse-21 subset is used for simple scenes and the Bedroom-28 subset is used for complex scenes. We mark the better results as **bold** font.

| Method | Simple Scene | Complex Scene |
|---|---|---|
| Generic Solution | 63.34 | **47.55** |
| Specific Solution | **63.70** | 45.41 |

## D.2 Quantitative Comparison Results

In this part, we present all the quantitative comparison results obtained following the protocol described in Section 5. These results are from a dataset of simple scenes (Horse-21 [2]) and a dataset of complex scenes (Bedroom-28 [2]). Since SDXL and Playground v2 share the same U-Net architecture, their results are shown in the same tables. We display the results from the lowest resolution in Table 4 and the results from the middle resolution in Table 5. We have also done a quantitative comparison on SDv1.5, and the results are shown in Table 6.

Table 4: Quantitative comparison results of SDXL and Playground v2. This table shows the results from the lowest resolution. Activation ID indicates the location of each activation in the diffusion U-Net. The best results are in **bold** font and the runner-up is underlined.

| Activation ID | SDXL | | Playground v2 | |
| --- | --- | --- | --- | --- |
| | Simple | Complex | Simple | Complex |
| up-level0-repeat0-res-out | 53.49 | 38.52 | 54.57 | 40.73 |
| up-level0-repeat0-vit-block0-cross-q | 56.46 | 42.31 | 55.32 | 41.93 |
| up-level0-repeat0-vit-block0-out | 57.36 | 42.31 | 56.60 | 44.52 |
| up-level0-repeat0-vit-block1-cross-q | 57.82 | 42.72 | 56.44 | 43.93 |
| up-level0-repeat0-vit-block1-out | 58.73 | 42.62 | 58.12 | 45.82 |
| up-level0-repeat0-vit-block3-cross-q | 58.27 | 43.09 | 57.79 | 45.90 |
| up-level0-repeat0-vit-block3-out | **58.95** | 44.83 | **59.04** | 47.81 |
| up-level0-repeat0-vit-block5-cross-q | 57.30 | 44.36 | 56.97 | 47.38 |
| up-level0-repeat0-vit-block5-out | 58.70 | **45.26** | 58.67 | **49.77** |
| up-level0-repeat0-vit-block7-cross-q | 57.24 | 43.00 | 57.69 | 48.51 |
| up-level0-repeat0-vit-block7-out | **58.95** | 44.83 | 58.76 | 48.39 |
| up-level0-repeat0-vit-block9-cross-q | 57.06 | 41.73 | 56.03 | 44.68 |
| up-level0-repeat0-vit-block9-out | 58.46 | 43.98 | 56.59 | 48.54 |
| up-level0-repeat0-vit-out | 58.36 | 41.54 | 57.99 | 43.95 |
| up-level0-repeat1-res-out | 57.28 | 39.27 | 56.59 | 41.74 |
| up-level0-repeat1-vit-block0-cross-q | 55.72 | 40.96 | 55.56 | 43.97 |
| up-level0-repeat1-vit-block0-out | 56.70 | 40.97 | 57.51 | 43.35 |
| up-level0-repeat1-vit-block1-cross-q | 57.92 | 41.65 | 56.37 | 42.10 |
| up-level0-repeat1-vit-block1-out | 57.38 | 42.25 | 57.20 | 43.10 |
| up-level0-repeat1-vit-block3-cross-q | 56.55 | 40.50 | 55.00 | 42.50 |
| up-level0-repeat1-vit-block3-out | 56.41 | 40.97 | 57.54 | 42.92 |
| up-level0-repeat1-vit-block5-cross-q | 54.92 | 39.66 | 54.89 | 40.79 |
| up-level0-repeat1-vit-block5-out | 55.86 | 40.66 | 56.32 | 42.48 |
| up-level0-repeat1-vit-block7-cross-q | 51.63 | 39.01 | 52.72 | 38.37 |
| up-level0-repeat1-vit-block7-out | 53.97 | 39.79 | 55.90 | 41.34 |
| up-level0-repeat1-vit-block9-cross-q | 50.27 | 36.42 | 36.09 | 24.81 |
| up-level0-repeat1-vit-block9-out | 53.31 | 38.58 | 53.19 | 40.30 |
| up-level0-repeat1-vit-out | 57.89 | 38.91 | 57.00 | 42.87 |
| up-level0-repeat2-res-out | 56.11 | 36.37 | 55.95 | 40.62 |
| up-level0-repeat2-vit-block0-cross-q | 56.09 | 36.68 | 55.56 | 41.29 |
| up-level0-repeat2-vit-block0-out | 57.14 | 37.16 | 56.48 | 41.14 |
| up-level0-repeat2-vit-block1-cross-q | 55.38 | 34.87 | 56.15 | 39.78 |
| up-level0-repeat2-vit-block1-out | 56.36 | 35.63 | 56.71 | 40.93 |
| up-level0-repeat2-vit-block3-cross-q | 55.26 | 34.40 | 54.98 | 38.70 |
| up-level0-repeat2-vit-block3-out | 55.50 | 34.61 | 56.77 | 39.36 |
| up-level0-repeat2-vit-block5-cross-q | 52.70 | 33.16 | 54.70 | 37.93 |
| up-level0-repeat2-vit-block5-out | 54.68 | 33.75 | 55.75 | 38.95 |
| up-level0-repeat2-vit-block7-cross-q | 51.91 | 31.97 | 52.92 | 35.58 |
| up-level0-repeat2-vit-block7-out | 53.07 | 32.43 | 54.07 | 36.91 |
| up-level0-repeat2-vit-block9-cross-q | 49.22 | 29.34 | 41.07 | 18.48 |
| up-level0-repeat2-vit-block9-out | 51.41 | 30.64 | 51.31 | 35.64 |
| up-level0-repeat2-vit-out | 53.77 | 33.79 | 54.24 | 37.53 |
| up-level0-upsampler-out | 53.21 | 32.85 | 53.97 | 36.30 |

Table 5: Quantitative comparison results of SDXL and Playground v2. This table shows the results from the middle resolution. Activation ID indicates the location of each activation in the diffusion U-Net. The best results are in **bold** font and the runner-up is underlined.

| Activation ID | SDXL | | Playground v2 | |
| --- | --- | --- | --- | --- |
| | Simple | Complex | Simple | Complex |
| up-level1-repeat0-res-out | 47.84 | 30.12 | 47.49 | 31.63 |
| up-level1-repeat0-vit-block0-cross-q | **49.08** | **31.75** | **47.96** | **32.53** |
| up-level1-repeat0-vit-block0-out | 46.74 | 31.14 | 46.49 | 30.71 |
| up-level1-repeat0-vit-block1-cross-q | 44.78 | 30.56 | 44.21 | 28.58 |
| up-level1-repeat0-vit-block1-out | 42.20 | 27.32 | 40.90 | 27.13 |
| up-level1-repeat0-vit-out | 46.93 | 29.35 | 46.08 | 30.15 |
| up-level1-repeat1-res-out | 41.36 | 26.81 | 41.51 | 28.09 |
| up-level1-repeat1-vit-block0-cross-q | 39.25 | 27.02 | 39.92 | 27.72 |
| up-level1-repeat1-vit-block0-out | 38.64 | 25.68 | 39.47 | 27.60 |
| up-level1-repeat1-vit-block1-cross-q | 36.76 | 24.57 | 37.20 | 25.82 |
| up-level1-repeat1-vit-block1-out | 34.74 | 23.20 | 34.88 | 24.32 |
| up-level1-repeat1-vit-out | 39.16 | 25.48 | 39.87 | 26.82 |
| up-level1-repeat2-res-out | 33.58 | 23.99 | 36.30 | 25.26 |
| up-level1-repeat2-vit-block0-self-k | 31.63 | 22.98 | 33.97 | 23.88 |
| up-level1-repeat2-vit-block0-cross-q | 32.16 | 24.97 | 34.30 | 24.83 |
| up-level1-repeat2-vit-block0-out | 31.31 | 23.51 | 34.81 | 25.27 |
| up-level1-repeat2-vit-block1-self-k | 30.39 | 22.99 | 33.01 | 25.44 |
| up-level1-repeat2-vit-block1-cross-q | 27.65 | 22.61 | 32.40 | 24.01 |
| up-level1-repeat2-vit-block1-out | 27.66 | 22.35 | 31.44 | 24.24 |
| up-level1-repeat2-vit-out | 27.61 | 19.80 | 26.29 | 18.61 |

Table 6: Quantitative comparison results of SDv1.5. Activation ID indicates the location of each activation in the diffusion U-Net. The best results are in **bold** font and the runner-up is underlined, both marked per resolution.

| Activation ID | SDv1.5 | |
| --- | --- | --- |
| | Simple | Complex |
| up-level1-repeat0-res-out | 47.16 | 37.86 |
| up-level1-repeat0-vit-block0-cross-q | 47.59 | 41.13 |
| up-level1-repeat0-vit-block0-out | 48.59 | 39.75 |
| up-level1-repeat0-vit-out | 48.21 | 39.91 |
| up-level1-repeat1-res-out | 49.71 | 40.80 |
| up-level1-repeat1-vit-block0-cross-q | 49.04 | **42.80** |
| up-level1-repeat1-vit-block0-out | 49.88 | 39.43 |
| up-level1-repeat1-vit-out | 50.23 | 40.98 |
| up-level1-repeat2-res-out | **50.61** | 39.47 |
| up-level1-repeat2-vit-block0-cross-q | 50.10 | 38.80 |
| up-level1-repeat2-vit-block0-out | 49.09 | 37.44 |
| up-level1-repeat2-vit-out | 49.62 | 36.69 |
| up-level2-repeat0-res-out | 52.43 | 37.98 |
| up-level2-repeat0-vit-block0-cross-q | 52.59 | 39.63 |
| up-level2-repeat0-vit-block0-out | 51.56 | 36.62 |
| up-level2-repeat0-vit-out | 52.29 | 38.78 |
| up-level2-repeat1-res-out | 53.30 | 36.65 |
| up-level2-repeat1-vit-block0-cross-q | **53.58** | **39.82** |
| up-level2-repeat1-vit-block0-out | 50.70 | 35.96 |
| up-level2-repeat1-vit-out | 50.50 | 35.59 |
| up-level2-repeat2-res-out | 48.47 | 33.70 |
| up-level2-repeat2-vit-block0-self-q | 45.44 | 32.45 |
| up-level2-repeat2-vit-block0-self-k | 45.69 | 32.71 |
| up-level2-repeat2-vit-block0-cross-q | 46.24 | 32.95 |
| up-level2-repeat2-vit-block0-out | 45.33 | 32.23 |
| up-level2-repeat2-vit-out | 45.04 | 28.87 |
| up-level2-upsampler-out | 45.07 | 29.57 |
| up-level3-repeat0-vit-block0-self-q | **38.81** | 25.72 |
| up-level3-repeat0-vit-block0-self-k | 37.78 | **26.04** |
| up-level3-repeat1-vit-block0-self-q | 31.64 | 24.02 |
| up-level3-repeat1-vit-block0-self-k | 32.07 | 24.00 |
| up-level3-repeat2-vit-block0-self-q | 29.24 | 21.27 |
| up-level3-repeat2-vit-block0-self-k | 29.19 | 21.39 |

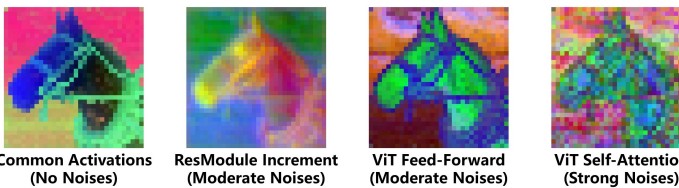

**Common Activations**
**(No Noises)**

**ResModule Increment**
**(Moderate Noises)**

**ViT Feed-Forward**
**(Moderate Noises)**

**ViT Self-Attention**
**(Strong Noises)**

Figure 16: This visualization compares the high-frequency noises in various activations, showing three types of increment activations with strong noises.

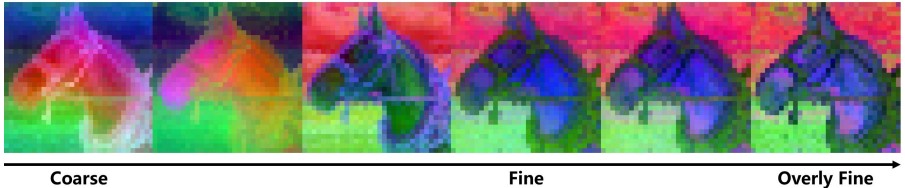

**Coarse**            **Fine**     **Overly Fine**

Figure 17: This visualization shows the granularity change across all inter-module activations of one resolution. At the end, the activations start to contain some slight noises, which is a sign of over-refinement.

## E  Other Properties of Diffusion U-Nets

### E.1  High-Frequency Noises in Increment Activations

Typically, high-frequency signals are usually noisy [5, 34]. This applies to the diffusion U-Net, and we can further examine what activations are more vulnerable to such noises. To be specific, the diffusion U-Net contains many residual connection structures, and their increment activations are high-frequency signals prone to noises. Such increment activations include:

 (i) The increment branch of ResModule. These activations are moderately noisy and thus usually less effective than inter-module activations.

 (ii) The feed-forward layer activations within ViTs. These activations are also moderately noisy. However, this results from their significantly more channels compared to other activations, which can reduce their noise magnitude.

(iii) The self-attention value activations within ViTs. These activations are severely noisy and suffer significant degradation as they are the nested inner increments within embedded ViTs.

These activations are visualized in Figure 16 for a clearer illustration. Additionally, the residual activations within ViTs are at the same time also increments to the main inter-module residual. However, these activations are not obviously affected by high-frequency noise, possibly due to their dual role as ViT residuals.

### E.2  Detailed In-Resolution Granularity Change

We have previously described the existence of in-resolution granularity changes, and this section will further detail the pattern of one such change. In one resolution of the up-stage, activations gradually shift from coarse to fine granularity, which aligns with intuition. However, the diffusion U-Net tends to overly refine the inter-module activations, resulting in slight noises in the last few inter-module activations due to excessive detail. This can be observed from the visualization of Figure 17. Consequently, a drop in discriminative performance is often seen near the end of one resolution.

### E.3  Collaboration between Embedded ViTs

Multiple ViTs exist within one resolution; for example, one resolution in the up-stage typically contains three ViTs. These ViTs collaborate in refining the main inter-module residual. During this process, each ViT exhibits an inner granularity change as it produces the increment activation, and the

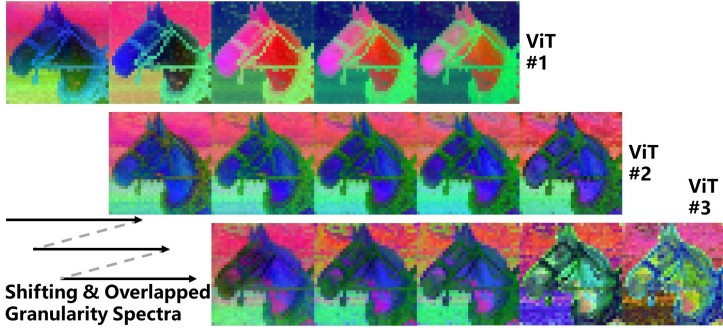

Figure 18: This visualization shows that the collaborating ViTs within one resolution contain overlapped and shifting granularity changes.

changes of collaborating ViTs form a certain pattern. To be specific, the change of one ViT overlaps with the previous one to some extent but also shifts as a whole towards finer granularity, which is shown in Figure 18.

## F    Future Direction

It might be a good future direction to focus on more challenging discrimination scenarios. Specifically, long tail and out-of-distribution are important problems in discrimination, which can greatly hinder the performance of models that work well under i.i.d. settings [49]. There have been a few attempts to address this more challenging problem with diffusion models. For example, the disentanglement property of prompts might grant diffusion features cross-domain capability [16]. It is also possible to utilize diffusion models to synthesize training samples to adjust training data distribution [50, 55]. Given these attempts, more efforts are still required to be put in this direction. Moreover, there is a recent study [17] that might boost long tail and out-of-distribution discrimination using diffusion models. To be specific, AUC is an evaluation metric as well as a loss function that promotes good performance on both head and tail samples, which is an important tool for long tail and out-of-distribution studies [47, 48]. Conventionally, AUC is applicable to image classification but not semantic segmentation and other pixel-level tasks, but the aforementioned study manages to adopt AUC for semantic segmentation. Given this new tool, it is now made more viable to attempt to enhance diffusion feature on long tail and out-of-distribution problems.

## G    Computation Resources

We use Nvidia(R) RTX 3090 and Nvidia(R) RTX 4090 GPUs for the experiments, all with 24GB VRAM. Most of our experiments, except label-scarce segmentation, require no additional storage besides the necessary space for model checkpoints and datasets. The label-scarce segmentation task first extracts features and stores them on the disk, and then loads them for the downstream task, which takes about 4GB.

The codes are designed to be able to run on a single GPU or less than 4 GPUs, while multiple experiments can run simultaneously if more GPUs are provided. Each experiment on the semantic correspondence task takes about 5 hours. Each experiment on the large-scale semantic segmentation task takes 2 to 3 days, depending on the dataset. Each experiment on the label-scarce segmentation task takes about 1 hour, but we repeat on 5 random splits following [2], which increases the overall time. All the experiments in sum can be done within two weeks.

Our quantitative comparison is based on the label-scarce segmentation task, each run taking about 40 minutes. The time is shorter because the quantitative comparison evaluates each activation individually. This comparison uses large storage, which is about 45GB per model per dataset. With the help of qualitative filtering, we can finish the quantitative comparison of one model on one dataset within 2 days.

# H Asset License

- SPair-71k: Available at https://cvlab.postech.ac.kr/research/SPair-71k/.
- Label-scarce segmentation datasets (sourced from LSUN): Available at https://github.com/fyu/lsun.
- ADE20K: Custom (research-only, non-commercial), at https://groups.csail.mit.edu/vision/datasets/ADE20K/terms/.
- CityScapes: Custom, at https://www.cityscapes-dataset.com/license/.

