# OpenReview forum: "Not All Diffusion Model Activations Have Been Evaluated as Discriminative Features"
_NeurIPS.cc/2024/Conference — NeurIPS 2024 spotlight_

### Official Review · Reviewer_xf1j · 2024-07-08

**Soundness:** 4
**Presentation:** 3
**Contribution:** 3
**Rating:** 7
**Confidence:** 5

**Summary:**

The paper addresses the critical task of enhancing feature selection in diffusion models to improve performance in discriminative tasks such as semantic correspondence, semantic segmentation, and label-scarce segmentation. Previous methods often overlooked many potential activations within diffusion models, leading to suboptimal performance and limitations due to ignoring high-resolution activations and not effectively handling diffusion noises. The authors propose a comprehensive feature selection solution that leverages distinctive properties of diffusion U-Nets, including diffusion noises, in-resolution granularity changes, and locality without positional embeddings, to filter and select the most relevant features. Experimental results demonstrate that their method outperforms state-of-the-art techniques, achieving significant improvements in various metrics across multiple datasets.

**Strengths:**

- This paper introduces a novel approach to feature selection in diffusion models, focusing on qualitative analysis to filter out suboptimal activations before performing quantitative comparisons. This methodology shifts from the traditional full-scale quantitative comparison, making the process more efficient and potentially more accurate.
- The authors identify three distinct properties of diffusion U-Nets that are leveraged for feature selection:
1.  **Asymmetric Diffusion Noises**: The diffusion process introduces unique noises affecting both low- and high-frequency signals.
2.  **In-Resolution Granularity Changes**: Modern diffusion U-Nets exhibit significant granularity changes within a single resolution due to fewer but larger resolutions.
3. **Locality without Positional Embeddings**: Self-attention modules in diffusion U-Nets show a new type of locality that enhances activation quality without traditional positional embeddings.
- The discovered properties means that the findings can be generalized to various diffusion models, providing valuable insights and a solid foundation for future research. This can guide the development of more effective and efficient discriminative models and applications in other fields.
- The proposed method achieves SOTA performance across multiple discriminative tasks, including semantic correspondence, semantic segmentation, and label-scarce segmentation.

**Weaknesses:**

- Some more comparisons are recommended. While the authors claim to find unique properties in diffusion U-Nets, it lacks a detailed comparison with traditional U-Nets. A more comprehensive analysis of how these properties differ would provide better context.
- Some visualizations, such as Figure 3(c) showing positional information in self-attention activations, could be clearer and better explained to enhance understanding of the findings.

**Questions:**

See the weakness.

**Limitations:**

Yes

---

> ### Author Rebuttal · Authors · 2024-08-06
>
> Thanks for your constructive comments, and we would like to make the following response.
>
> > **Weakness 1:**
> Some more comparisons are recommended. While the authors claim to find unique properties in diffusion U-Nets, it lacks a detailed comparison with traditional U-Nets. A more comprehensive analysis of how these properties differ would provide better context.
>
> **Response:**
> We will add more comparisons against traditional U-Nets during refinement, and here we provide some details.
>
> - Diffusion noises are a special noise type induced by the diffusion process, which inputs partially noisy images to diffusion U-Nets and expects the outputs to be noises. In traditional U-Nets without diffusion process, such as a U-Net for semantic segmentation, the input is image and the output is semantic masks. Therefore, there are no special diffusion noises in the activations of a traditional U-Net.
> - In-resolution granularity change, in theory, also exists in traditional U-Nets. However, in most traditional U-Nets, one resolution only has much fewer network components. In this way, in-resolution granularity change can hardly be observed.
> - Locality without positional embeddings is in fact a unique property compared to typical ViTs instead of traditional U-Nets. Traditional U-Nets are typically comprised of convolutional layers rather than attention layers, so neither locality nor positional information can be seen in traditional U-Nets.
>
> Additionally, we provide activation visualization from a traditional U-Net for semantic segmentation in the PDF attached to the global rebuttal.
>
> > **Weakness 2:**
> Some visualizations, such as Figure 3(c) showing positional information in self-attention activations, could be clearer and better explained to enhance understanding of the findings.
>
> **Response:**
> We will pay attention to this advice during refinement, and here are some changes we can make.
>
> The visualization provided throughout the paper is obtained using PCA analysis, reducing the dimension of activations down to 3 and regarding the three dimensions as RGB. Hence, the color of a latent pixel in the visualization can reflect the original information in the activation to some extent. Observe the visualization with an orange circle mark in Figure 3(c), and we can see that the latent pixel on the horse's neck is a light blue color, similar to the pixels to its left that actually represent the background. In contrast, the pixels above the circle are in purple color, though they also represent the horse's neck. Such comparison shows that a latent pixel is more similar to other pixels that are spatially near it than those semantically closer to it, which is the meaning of positional information and locality.

---

### Official Review · Reviewer_8YQr · 2024-07-11

**Soundness:** 4
**Presentation:** 3
**Contribution:** 3
**Rating:** 7
**Confidence:** 5

**Summary:**

This paper highlights the importance of considering a broader range of activations within diffusion models. The authors propose three universal properties of diffusion U-Nets that aid in qualitatively filtering out activations that are clearly sub-optimal. On top of this, the authors can improve the efficiency of feature selection. By leveraging these properties, the proposed method demonstrates superior performance across multiple discriminative tasks, such as semantic correspondence and segmentations. This comprehensive approach to activation selection addresses a fundamental issue in diffusion models, enhancing their applicability and performance in various tasks.

**Strengths:**

1) The authors propose an efficient method for selecting high-quality activations.  With the help of qualitatively filtering out sub-optimal activations before quantitative comparison, the computational costs can be significantly reduced, compared with the prior arts.
2) The authors present their observations in a clear and organized manner, making it easy for readers to understand the complex concepts and follow the research process. Besides, the main content and the appendix include numerous visual samples and activation visualizations, which help illustrate the properties and effectiveness of the proposed activation selection method.
3) Extensive experimental results across multiple discriminative tasks validate the superiority of the proposed method over the competitors.

**Weaknesses:**

W1: This paper could benefit from introducing more notations to clearly define and differentiate between various activations and components within the diffusion models, which would further enhance the clarity of the descriptions.
W2: While the empirical results are robust, the paper could include more theoretical analysis to provide deeper insights.
W3: Some more latest related work are commended to be cited:
[1] Diffusion 3D Features (Diff3F): Decorating Untextured Shapes with Distilled Semantic Features, CVPR 2024. [2] DiffSeg: Towards Detecting Diffusion-Based Inpainting Attacks Using Multi-Feature Segmentation, CVPR 2024.

**Questions:**

I have no more questions.

**Limitations:**

Yes

---

> ### Author Rebuttal · Authors · 2024-08-06
>
> Thanks for your constructive comments, and we would like to make the following response.
>
> > **W1:**
> This paper could benefit from introducing more notations to clearly define and differentiate between various activations and components within the diffusion models, which would further enhance the clarity of the descriptions.
>
> **Response:**
> Thanks for this good advice. We do use some notations in our codes, but we wanted to make the manuscript look more formal, so the notations were not used in the paper's main body. During refinement, we will try to add these notations back. For example, we use "up-resolution0-vit0-block1-cross-q" to represent the activation at cross-attention query, the second basic block, the first ViT, the first resolution, up-stage.
>
> > **W2:**
> While the empirical results are robust, the paper could include more theoretical analysis to provide deeper insights.
>
> **Response:**
> Thanks for your advice! We follow the fellow studies in this direction to take a utilitarian approach and leave more theoretical analysis to future work.
> For more details, please refer to the **global response Question 2**.
>
> > **W3:**
> Some more latest related work are commended to be cited.
>
> **Response:**
> Thanks for the advice. These studies will be added to the refined manuscript.
>
> - [1] is an application of diffusion features, which generates images conditioned on sampled views of 3D models, extracts diffusion features during generation, and unprojects features back to the 3D surface. This work demonstrates that even though features from different views can be inconsistent, the associated features are robust.
> - [2] detects the areas having been inpainted in an image based on segmentation using multiple features, including RGB-based, frequency-based, and noise-based features.

---

### Official Review · Reviewer_cCcb · 2024-07-13

**Soundness:** 3
**Presentation:** 3
**Contribution:** 4
**Rating:** 7
**Confidence:** 5

**Summary:**

Diffusion models have achieved significant success in image generation and show great potential for various discriminative tasks. The authors rethink the foundational problem of feature selection within these models. To this end, they analyze the properties of diffusion models, including asymmetric diffusion noises, in-resolution granularity changes, and locality without positional embeddings, to filter out suboptimal activations. The results demonstrate that their proposed feature selection method outperforms state-of-the-art techniques across multiple discriminative tasks, achieving superior performances.

**Strengths:**

- This paper points out three unique properties of diffusion models—asymmetric diffusion noises, in-resolution granularity changes, and locality without positional embeddings—that provide valuable insights into improving discriminative tasks.
- The authors propose an off-the-shelf solution for feature selection. Besides, the findings and methodologies presented are not limited to the specific models studied (SDv1.5 and SDXL).
- The paper conducts extensive experiments across multiple discriminative tasks, including semantic correspondence, semantic segmentation, and label-scarce segmentation. The results, as well as the visual illustration, validate the effectiveness of the proposed feature selection method, demonstrating significant improvements over state-of-the-art techniques.

**Weaknesses:**

-  The authors acknowledge the uncertainty about whether the findings can generalize to newer models like DiT (Diffusion Transformer) due to architectural differences.
- The qualitative filtering approach proposed is novel, but its scalability and efficiency in very large-scale settings are not fully demonstrated. How do the authors alleviate this problem?
- The paper focuses on empirical results and qualitative analysis. There is a lack of rigorous theoretical foundation or mathematical proofs to support the proposed feature selection methodology and the identified properties of diffusion U-Nets. A more robust theoretical framework would strengthen the claims made.

**Questions:**

Besides those in weakness, I wonder can we combine multiple features during the qualitative analysis to further improve the performance? And how many features do you select for each image?

**Limitations:**

Yes, as stated in Sec.7.

---

> ### Author Rebuttal · Authors · 2024-08-06
>
> Thanks for your constructive comments, and we would like to make the following response.
>
> > **Weakness 1:**
> The authors acknowledge the uncertainty about whether the findings can generalize to newer models like DiT (Diffusion Transformer) due to architectural differences.
>
> **Response:**
> Thanks for this valuable suggestion! There are currently still many researchers and casual users using U-Net-based diffusion models. We will leave the study on DiT models to future work.
> For more details, please refer to the **global response Question 1**.
>
> > **Weakness 2:**
> The qualitative filtering approach proposed is novel, but its scalability and efficiency in very large-scale settings are not fully demonstrated. How do the authors alleviate this problem?
>
> **Response:**
> The qualitative filtering is mainly conducted through feature visualization. With the prior knowledge of humans, it is relatively easy to do sampling, i.e., only visualizing some activations, for better efficiency. Furthermore, the three discovered properties can help this process be more efficient. More practically, SDXL is one of the current largest diffusion models, and the qualitative filtering for it has been carried out successfully.
>
> > **Weakness 3:**
> The paper focuses on empirical results and qualitative analysis. There is a lack of rigorous theoretical foundation or mathematical proofs to support the proposed feature selection methodology and the identified properties of diffusion U-Nets. A more robust theoretical framework would strengthen the claims made.
>
> **Response:**
> We follow the utilitarian trend of fellow studies and leave more rigorous theoretical analysis to future work.
> For more details, please refer to the **global response Question 2**.
>
> > **Question 1:**
> I wonder can we combine multiple features during the qualitative analysis to further improve the performance?
>
> **Response:**
> The major benefit of combining multiple features is that different features might contain complementary information. Therefore, it is better to combine features that are more distinct. If we aim to do this via qualitative analysis, one possible way is to select activations with as distinct color patterns as possible. Note that the colors we can see in the visualization are calculated using PCA analysis with the target dimension as 3, so the colors can reflect the information contained by activations to some extent.
>
> > **Question 2:**
> And how many features do you select for each image?
>
> **Response:**
> Except for the Ours-XL-t solution, we select four features per image. This is to ensure the total dimension of features is almost the same as the conventional features [2, 41]. To be specific, the conventional features in our experiment have 3520 dimensions, Ours-v1.5 features have 3520 dimensions, and Ours-XL features have 3840 dimensions. Ours-XL-t does select more features to match the feature amalgamation technique, where 10 or 8 features in total are selected based on the task.

---

### Official Review · Reviewer_AMWL · 2024-07-14

**Soundness:** 3
**Presentation:** 3
**Contribution:** 3
**Rating:** 7
**Confidence:** 4

**Summary:**

This paper proposes a new feature selection method for diffusion models by evaluating a broader range of activations, particularly those in embedded Vision Transformers (ViTs). The authors identify the limitations of current approaches that consider only a narrow range of activations and introduce a qualitative analysis to filter out low-quality activations in diffusion U-Nets. They develop specific feature selection solutions for popular diffusion models SDv1.5 and SDXL. Experiments demonstrate that this method outperforms state-of-the-art techniques in tasks like semantic correspondence, semantic segmentation, and label-scarce segmentation, showcasing its effectiveness and generalizability.

**Strengths:**

- This paper extends the evaluation to a wider range of activations within diffusion models, especially those in embedded ViT modules, which were previously overlooked. Since feature selection is a basic problem for diffusion features, this paper can boost future work in this direction.
- The authors introduce a qualitative analysis to effectively filter out low-quality activations, simplifying the subsequent quantitative comparison and improving feature selection efficiency. What’s more, the observed properties are not limited to the discussed models, which can also inspire future work.
- Extensive experiments demonstrate that the proposed feature selection method significantly outperforms state-of-the-art techniques in various discriminative tasks, validating its effectiveness and generalizability.

**Weaknesses:**

- As pointed out by the authors, the observations and methods proposed do not necessarily generalize well to recently developed DiT models since their architecture is markedly different from diffusion U-Nets .
- Some details can be more clarified. For example, in Appenidx C.3, the authors state that “this is also the setting for the quantitative comparison”. In other words, quantitative comparison is conducted on Label-Scarce Segmentation, and the experiments on the other tasks follow this setting. Is this understanding correct? If so, I wonder why the authors select this setting to conduct quantitative comparison.
- Besides, there are some typos. For example, in the introduction, “a fundamental problem to select” should be “a fundamental problem in selecting”. In the caption of Figure 3, “existing knowledge of other models” should be “existing knowledge about other models”.

**Questions:**

Please refer to the weakness.

**Limitations:**

Yes, the authors have clarified the limitations in Section 7.

---

> ### Author Rebuttal · Authors · 2024-08-06
>
> Thanks for your constructive comments, and we would like to make the following response.
>
> > **Weakness 1**:
> As pointed out by the authors, the observations and methods proposed do not necessarily generalize well to recently developed DiT models since their architecture is markedly different from diffusion U-Nets.
>
> **Response:**
> Thanks for this valuable suggestion! Despite this limitation, U-Net-based diffusion models are still being vastly used, so our study can have a broad impact.
> For more details, please refer to the **global response Question 1**.
>
> > **Weakness 2:**
> Some details can be more clarified. For example, in Appendix C.3, the authors state that “this is also the setting for the quantitative comparison”. In other words, quantitative comparison is conducted on Label-Scarce Segmentation, and the experiments on the other tasks follow this setting. Is this understanding correct? If so, I wonder why the authors select this setting to conduct quantitative comparison.
>
> **Response:**
> Thanks for the advice. We will try to better clarify details in the refined manuscript. As for the specific question, the reviewer's understanding is correct. We choose to conduct the quantitative comparison on label-scarce segmentation because (i) Experiments on this task take a relatively short time. (ii) This task provides several datasets for scenes of different complexity, allowing us to compare the capability of activations in different scenarios.
>
> > **Weakness 3:**
> Besides, there are some typos.
>
> **Response:**
> Thanks for the advice. We will pay attention to such typos and fix them in the revised manuscript.

---

### Author Rebuttal · Authors · 2024-08-06

Dear SAC, AC, and reviewers,

Thank you for your invaluable feedback. Based on your comments, we have revised the details and now offer a global response to some common questions.

> **Question 1:**
As we have admitted, the conclusion of this study can fail to extend to DiT models. Can this be an important weakness?

**Response:**
Despite the advancements in DiT models, the more conventional U-Net-based diffusion models are still being vastly used by both casual users and researchers. Hence, our study can still benefit many fellow studies.
Moreover, the study on DiT features may require a dedicated study, so we narrow the scope of this paper to more concentrated research.

> **Question 2:**
This study takes a more empirical and qualitative approach, without rigorous theoretical foundation or mathematical proofs.

**Response:**
Theoretical analysis can indeed bring more insights, and we will set this as our future plan. Nevertheless, most fellow studies in this direction take a rather utilitarian approach, placing enhancing the actual performance in the first place. We also follow this trend and argue that the current manuscript has achieved this goal.

---

Please refer to the specific responses below for more information. We will update all these improvements in the next version.

---

### Decision · Program_Chairs · 2024-09-25

**Decision:**

Accept (spotlight)

**Comment:**

The paper considers the problem of feature selection within diffusion models, analyzing various properties of diffusion models (properties such as asymmetric diffusion noises, in-resolution granularity changes, and locality without positional embeddings), and proposing a feature selection method that outperforms state-of-the-art techniques across multiple tasks.  Empirical results are
detailed and validate the claims.  All reviewers agree with the4 value of the contribution.